# Potential of Plant-Derived Compounds in Preventing and Reversing Organ Fibrosis and the Underlying Mechanisms

**DOI:** 10.3390/cells13050421

**Published:** 2024-02-28

**Authors:** Patrícia dos Santos Azeredo, Daping Fan, E. Angela Murphy, Wayne E. Carver

**Affiliations:** 1Laboratory of Atherosclerosis, Thrombosis and Cell Therapy, Institute of Biology, State University of Campinas—UNICAMP Campinas, Campinas 13083-970, Brazil; p264504@dac.unicamp.br; 2Department of Cell Biology and Anatomy, School of Medicine, University of South Carolina, Columbia, SC 29209, USA; daping.fan@uscmed.sc.edu; 3Department of Pathology, Microbiology and Immunology, School of Medicine, University of South Carolina, Columbia, SC 29209, USA; angela.murphy@uscmed.sc.edu

**Keywords:** fibrosis, plant compounds, alternative medicine, myofibroblast, mechanism

## Abstract

Increased production of extracellular matrix is a necessary response to tissue damage and stress. In a normal healing process, the increase in extracellular matrix is transient. In some instances; however, the increase in extracellular matrix can persist as fibrosis, leading to deleterious alterations in organ structure, biomechanical properties, and function. Indeed, fibrosis is now appreciated to be an important cause of mortality and morbidity. Extensive research has illustrated that fibrosis can be slowed, arrested or even reversed; however, few drugs have been approved specifically for anti-fibrotic treatment. This is in part due to the complex pathways responsible for fibrogenesis and the undesirable side effects of drugs targeting these pathways. Natural products have been utilized for thousands of years as a major component of traditional medicine and currently account for almost one-third of drugs used clinically worldwide. A variety of plant-derived compounds have been demonstrated to have preventative or even reversal effects on fibrosis. This review will discuss the effects and the underlying mechanisms of some of the major plant-derived compounds that have been identified to impact fibrosis.

## 1. Introduction

The extracellular matrix (ECM) is a dynamic network of secreted and cell-bound macromolecules including fibrillar collagens, non-collagenous glycoproteins, proteoglycans, and other components. The ECM plays many roles in tissues including functioning as a three-dimensional scaffold essential for the development and maintenance of organ structure. Interactions between cells and their surrounding ECM have been shown to impact many cellular processes including proliferation, migration, differentiation and even survival as reviewed by Theocharis and colleagues [1]. Alterations in ECM composition, organization, or accumulation can deleteriously impact embryonic development and organ homeostasis in adults. For instance, deficient collagen α(1)I expression in mice (Mov-13 mouse), leads to vascular fragility and the formation of aneurysms [2]. Mutations have been identified in over 100 ECM components that result in congenital defects including congenital muscular dystrophy (laminin-211 α2 chain), Marfan syndrome (fibrillin 1), osteogenesis imperfecta (collagen type I α1 and α2 chains) and others (reviewed in [3]). In the mature organ, excessive accumulation of ECM components or fibrosis often results in organ dysfunction.

Fibrosis is an outcome of dysregulated tissue repair in response to stress or organ damage. Accumulation of excessive ECM components is in fact a normal and necessary but transient event in tissue repair. When repair is successful, the increase in ECM resolves. However, when this process is dysregulated, the accumulation of ECM continues, leading to alterations in organ architecture, biomechanical properties, and function. Fibrosis can affect any organ and can be initiated by a wide range of stresses. As such, it contributes extensively to mortality and morbidity [4], contributing up to 35.4% in 2019 of global deaths, depending on the classification of fibrosis [5]. The realization of the impact of alterations in the ECM on organ function and technological innovations in ECM research has greatly expanded our understanding of the underlying mechanisms of fibrosis (reviewed in [4]). Understanding the regulation of fibrosis has become increasingly important as it is now realized that in some settings fibrosis is reversible, at least in its early stages (reviewed in [6,7,8]).

While fibrosis is a complex process and ultimate patient outcomes are dependent on the organ(s) involved, a number of similarities in the fibrosis process have been identified across organs. Fibrosis is initiated by a wide range of stressors including toxins and physical trauma [9,10,11]. This generally results in an inflammatory response, particularly involving resident inflammatory cells including macrophages and mast cells. Inflammatory mediators released by these cells result in part in the activation of mesenchymal cells including resident fibroblasts into a myofibroblast phenotype that produce ECM components and additional cytokines, chemokines and growth factors [12,13]. These responses are designed to repair/regenerate damaged tissue and resolve the stress response. Prolonged activation of this response can lead to cellular destruction resulting in continued activation of inflammatory cells and myofibroblasts and perpetuation of the fibrotic response [14].

While a number of secreted factors promote the fibrotic response including connective tissue growth factor (CTGF), platelet-derived growth factors and select interleukins; transforming growth factor-beta (TGF-β) isoforms, particularly TGF-β1, play central roles in this process (Figure 1). For example, extensive evidence from isolated cells and animal models indicates that activation of TGF-β canonical (Smad 2/3) and non-canonical pathways enhance expression and accumulation of ECM components [15,16] and promote myofibroblast formation [17]. In many cases, the levels of TGF-β correlate to the degree of fibrosis in animal models and patients [18,19] and this cytokine is being extensively studied as a potential noninvasive biomarker of fibrosis progression. Furthermore, blockade of TGF-β or its downstream signaling pathways is sufficient to prevent fibrogenesis [20,21]. Not surprisingly, many inhibitors of fibrosis including some plant-derived compounds (discussed below) directly or indirectly target TGF-β or its signaling pathways.

A key step in the progression of fibrosis is the activation of ECM-producing cells into a myofibroblast or myofibroblast-like phenotype. These cells have been reviewed extensively elsewhere and will only be briefly mentioned here [22,23]. Myofibroblasts are generally characterized by enhanced contractile activity, formation of stress fibers, abundant synthesis of ECM components and expression of α-smooth muscle actin (α-SMA). In addition, these cells produce cytokines and growth factors that promote the fibrotic response in an autocrine/paracrine manner (reviewed in [24]). Myofibroblasts are derived from a variety of cells in response to tissue damage and stress including quiescent fibroblasts, blood-derived fibrocytes, mesenchymal stem cells, stellate cells of the liver and others [25,26]. Regardless of their origin, myofibroblasts likely arise as an acute and beneficial response that helps repair damaged tissue. Continued myofibroblast contraction and production of ECM components becomes deleterious and, in many cases, yields stiff fibrotic tissue that obstructs and negatively impacts organ function. As discussed below, a number of plant-derived compounds modulate the formation or survival of myofibroblasts via diverse mechanisms ultimately leading to the repair of damaged tissue.

Despite the substantial morbidity and mortality of fibrotic disease, a limited number of drugs have been approved by the U.S. Food and Drug Administration (FDA) specifically for anti-fibrotic treatment. Several hurdles exist in the development of anti-fibrotic treatments, including (1) the complexity of fibrotic signaling involving many interacting pathways that make in vivo responses difficult to predict and likely necessitate combinatory therapies to achieve adequate response, (2) undesirable side effects in non-diseased tissues, (3) slow progression of fibrotic buildup that necessitates long and costly clinical trials, and (4) highly variable patient responses that lead to statistically insignificant results even though some patients show improvement under treatment (reviewed in [27,28]). Clearly more effective treatments are needed for fibrosis and a number of studies have focused on the potential of plant-derived compounds as therapeutic agents for halting or reversing the fibrotic process given their broad pleiotropic effects. The effects of some of the major compounds that modulate fibrosis and their major underlying mechanisms are reviewed in the following discussion and summarized in Table 1 and Table 2.

## 2. Major Plant-Derived Compounds that Modulate Fibrosis

### 2.1. Curcumin

Diferuloylmethane, commonly known as curcumin, is a yellow pigment extracted from turmeric, a spice and coloring agent derived from *Curcuma longa* [29]. This plant-derived compound has demonstrated diverse roles within biological systems, encompassing anti-thrombotic, anti-inflammatory, antioxidant, anti-proliferative, chemopreventive, anti-angiogenic and pro-apoptotic effects (reviewed in [30,31]). Consequently, it and its active metabolites hold potential as therapeutic agents for various diseases. Furthermore, curcumin, administered orally or intravenously, has already been utilized in various medical conditions in humans (reviewed in [32]). The limited bioavailability of curcumin prompts the exploration of alternative approaches. To overcome this challenge, researchers have employed various modified versions of natural curcumin, along with synthetic analogs and derivatives [33].

Curcumin effectively suppresses TGF-β1-induced myofibroblast differentiation of fibroblasts [34,35]. These results align with research demonstrating that curcumin impacts fibrosis in isolated cells and animal models by reducing the expression of TGF-β1 [36]. The connective tissue growth factor (CTGF) is associated with various fibrotic conditions and is induced by TGF-β. Studies have demonstrated that curcumin can inhibit TGF-β1-induced CTGF expression by disrupting Smad2 signaling and can also suppress CTGF through the inhibition of the c-Jun NH(2)-terminal kinase (JNK) signaling pathway in fibroblasts [37,38]. In a rat model of peritoneal fibrosis, Zhao and colleagues elucidated that the inhibitory action of curcumin on TGF-β signaling is mediated, at least in part, through the suppression of the Transforming Growth Factor-Activated Kinase 1 (TAK1) pathway [39].

As mentioned in the Introduction, it is now clear that immune/inflammatory responses and fibrosis are closely integrated. Another mechanism through which curcumin inhibits fibrosis is by regulating macrophage–fibroblast crosstalk. Curcumin treatment promoted apoptosis of macrophages and inhibited lipopolysaccharide-induced secretion of pro-inflammatory cytokines by these cells [40]. In the same study, exposure to curcumin-treated macrophages also inhibited the pro-fibrotic response of fibroblasts to TGF-β in a co-culture model. This effect centered around attenuation of interleukin (IL)-18 secretion and Smad 2/3 activation in fibroblasts that were co-cultured with curcumin-treated macrophages compared to macrophages not exposed to curcumin. The response in the co-culture model was consistent with decreased serum levels of IL-18 and diminished Smad 2/3 activation in heart tissue in response to curcumin in a rat myocardial infarction model [40].

The angiotensin II (Ang II) pathway is closely associated with fibrosis (Figure 1). Ang II acts through multiple cell-surface receptors and its fibrotic effects appear to be primarily associated with the Ang II type 1 (AT1) receptor [41] while activation of the AT2 receptor may have anti-fibrotic effects [42]. However, this is likely an overly simplistic interpretation of a complex system and novel components of this system are continuing to be described [43]. In the heart, binding of the AT1 receptor promotes the proliferation and migration of fibroblasts, increases apoptosis in the myocardium and results in the accumulation of collagen I, III and fibronectin [44]. Additionally, angiotensin II upregulates plasminogen activator inhibitor-1 (PAI1), leading to an enhanced synthesis of ECM along with the inhibition of matrix metalloproteinases (MMPs) and a reduction in matrix degradation. These factors collectively contribute to the profibrotic environment (reviewed in [44]). Studies have demonstrated that the protein level of the AT1 receptor was reduced in animals treated with curcumin, while the AT2 receptor was up-regulated [45]. This resulted in an increased ratio of the AT2 receptor to the AT1 receptor in the curcumin group. Alongside these modulations, curcumin significantly decreased the populations of macrophages and α-SMA-expressing myofibroblasts. Collagen type I synthesis was inhibited, leading to attenuation of tissue fibrosis, as evidenced by less extensive collagen-rich ECM. In vitro, curcumin has been shown to attenuate the Ang II-induced inflammatory response in vascular smooth muscle cells [46]. The underlying mechanisms of this effect are, at least in part, due to the activation of peroxisome proliferator-activated receptor-γ (PPAR-γ), which is highly effective in diminishing inflammation and the generation of intracellular reactive oxygen species (ROS).

The family of serine/threonine protein kinases known as protein kinase C (PKC) includes multiple isozymes and their function is linked to pathological conditions including fibrosis [47]. PKC becomes activated when cytosolic Ca^2+^ levels rise, facilitated by its binding to diacylglycerol present in the cell membrane. This activation sets off various downstream signaling pathways, one of which is the mitogen-activated protein kinase (MAPK) pathway. This pathway contributes to a variety of intracellular effects, including the regulation of cell growth and proliferation. Curcumin has generally been thought to inhibit PKC activity; however, a study suggested that this is dependent on cellular calcium ion (Ca^2+^) concentrations and PKC localization [48]. That is, curcumin diminished the activity of PKC and its translocation to the cell membrane and mitigated downstream pathways when exposed to lower Ca^2+^ concentrations. However, it elicited an increase in PKC activity when Ca^2+^ concentrations were higher. Diabetic cardiomyopathy is accompanied by increased activation and membrane-association of PKC [49]. In a rat streptozotocin-induced model of diabetes, treatment with curcumin reduced the activation of the PKC α and β2 isoforms in left ventricular tissue [50]. Furthermore, treatment with curcumin attenuated the phosphorylation of p38 and extracellular signal-regulated kinase 1/2 (ERK) MAPKs in the heart in addition to decreasing the levels of TGF-β and ECM. This suggests that curcumin-mediated modulation of PKC-MAPK signaling may be an important component of curcumin’s anti-fibrotic effects (Figure 1).

### 2.2. Capsaicin

Capsaicin is the primary bioactive substance in chili pepper extract derived from the Capsicum genus (Table 2). It is a naturally occurring alkaloid characterized by its hydrophobic, crystalline, colorless and odorless properties. Studies demonstrated that capsaicin exhibits dose-dependent anti-cancer, analgesic and cardioprotective effects (reviewed in [51]). The “heat-sensation” induced by capsaicin arises from its binding to transient receptor potential vanilloid (TRPV) ion-channel receptors. However, certain biological activities of capsaicin, such as its anti-neoplastic and cardioprotective effects, have been identified as independent of the TRPV1 receptor [52].

Liu and colleagues utilized two mouse models to evaluate the effects of capsaicin on fibrosis associated with chronic kidney disease. Animals treated with intraperitoneal capsaicin showed significantly ameliorated renal fibrosis [53]. This effect was achieved, at least in part, by inhibiting the activation of myofibroblasts and protecting against the phenotypic alteration of tubular epithelial cells, primarily through the inhibition of TGF-β1/Smad2/3 signaling pathways. Another in vivo study utilizing carbon tetrachloride to induce liver fibrosis revealed that mice treated with capsaicin exhibited reduced liver fibrosis through the attenuation of the inflammatory response and macrophage M1 (i.e., pro-inflammatory phenotype) polarization [54]. This improvement was accompanied by reduced tumor necrosis factor alpha (TNF-alpha) levels in the serum, achieved by targeting Notch signaling.

Several pathways in addition to TGF-β/Smad 2/3 have been postulated to mediate the effects of capsaicin (Figure 1 and Table 1). Treatment of the human hepatoma cell line, HepG2, with capsaicin, resulted in suppression of the protein kinase B (Akt)/mammalian target of the rapamycin (mTOR) pathway and upregulation of PPAR-g protein expression [55]. Among other things, PPAR-g signaling mitigates oxidative stress and inflammation, and activation of this pathway has become a potential therapeutic strategy for fibrosis. Another important pathway in fibrosis that is targeted by several plant-derived compounds is the eukaryotic translation initiation factor 3a (eIF3a) pathway. eIF3a is a versatile protein with a pivotal role in the regulation of diverse cellular functions, such as proliferation and differentiation. Notably, the expression of eIF3a exhibited a pronounced increase in the lungs of rats afflicted with pulmonary fibrosis, coinciding with the up-regulation of α-SMA and collagens [56]. Results from a study conducted by Liu and colleagues using a bleomycin-induced pulmonary fibrosis mouse model illustrated that capsaicin at a modest dose for 21 days reversed mesenchymal transition of alveolar epithelial cells (epithelial-to-mesenchymal transition or EMT) and mitigated fibrosis [57]. This beneficial effect was associated with inhibition of the Erk 1/2/eIF3a signaling pathway.

Doxorubicin is an anthracycline drug widely used as a chemotherapeutic agent for certain cancer types. A well-known side effect of doxorubicin is its cardiotoxicity which includes the death of cardiomyocytes and myocardial fibrosis. A recent study illustrated that capsaicin attenuates the cardiotoxic effects of doxorubicin in a mouse model [58]. In these studies, capsaicin inhibited doxorubicin-induced myocardial fibrosis and cardiomyocyte apoptosis. The latter effect was mediated, at least in part, by capsaicin activation of the phosphoinositide 3-kinase (PI3K)—Akt signaling pathway. Capsaicin also attenuated cardiomyocyte ferroptosis in vivo and in vitro via maintenance of cellular iron levels. However, it was not clear whether capsaicin had a direct effect on ECM-producing cells or if reduced fibrosis was a consequence of attenuated cardiomyocyte death and cell death-related signaling to fibroblasts.

### 2.3. Ellagic Acid

Ellagic acid (EA) is present within ellagitannins, primarily found in vegetables, nuts, and fruits like raspberries and pomegranates ([59] and Table 2 herein). The formation of EA occurs through the hydrolysis of ellagitannins, which serve as secondary metabolites in plants. EA is notably recognized for its anti-proliferative and antioxidant properties [59,60,61]. Due to its low solubility and permeability, EA faces challenges in terms of oral bioavailability and practical applications in the clinic. To address the limitations associated with physicochemical parameters, various formulation techniques have been employed. These methods include particle size reduction, amorphization and the utilization of lipid-based formulations. Notably, certain approaches have demonstrated substantial enhancements in both the solubility and bioavailability of EA (reviewed in [62]).

A study conducted by Zhao and colleagues [63] in vitro using fibroblasts demonstrated that EA significantly reduced the expression of TGF-β1 and phosphorylation of Smad2 and Smad3, thereby preventing the activation of the canonical TGF-β/Smad2/3 signaling pathway. In addition, EA treatment decreased the viability and migration of fibroblasts and downregulated the expression of ECM components including collagen type I and fibronectin. In this way, EA attenuated the development of hypertrophic scars by restraining the survival and movement of hypertrophic scar fibroblasts and repression of the profibrotic gene expression program. In an in vivo study investigating the effects of EA on chronic renal failure in Wistar albino rats (5/6 nephrectomy model), a 16-week treatment with EA markedly reduced renal damage and collagen deposition [64]. EA also inhibited the expression of TGF-β and fibronectin in renal tissues and significantly decreased the serum levels of pro-inflammatory cytokines, including IL-1β, IL-6 and TNF-α. The anti-fibrotic effect is noteworthy, considering that pro-inflammatory cytokines such as TNF-α have been documented to induce the expression of TGF-β in fibroblasts [65].

Treatment with EA attenuated cardiac fibrosis and downregulated mRNA expression of fibrosis-related genes including collagens type I and III in a rat model of myocardial infarction [66]. These results were reproduced in vitro, with cultured cardiac fibroblasts treated with Ang II with or without EA. In this setting, EA also inhibited cell proliferation and migration stimulated by Ang II. Treatment with EA also dose-dependently depressed histone deacetylase (HDAC) 1 expression in isolated cardiac fibroblasts. Overexpression of HDAC 1 was able to counter the effects of EA, suggesting that modulation of HDAC 1 plays a role in the effects of EA on fibrosis. In addition, EA has been shown to potently inhibit angiotensin converting enzyme 1 (ACE 1) activity, the primary mechanism for the conversion of Ang I to Ang II [67], which may provide another mechanism whereby EA inhibits fibrosis in vivo.

A study conducted by Baradaran and colleagues revealed that the protective effects of low-dose EA on aging in a mouse model were entirely nullified by a PPAR-γ antagonist, GW9662 [68]. This finding suggests that EA partly exerts its effects through activation of PPAR-γ. Though there are some discrepancies in the literature, EA has also been identified to downregulate the expression of classical PKC isozymes, specifically PKCα, PKCβ and PKCγ [69]. Oxidized low-density lipoprotein (oxLDL) induces the expression of MMPs in vascular endothelial cells. Studies with human umbilical vein endothelial cells (HUVECs) illustrated that EA can inhibit oxLDL-induced expression of MMP-1 and MMP-3 and this effect is achieved by modulating the PKCα/PPAR-γ/Nuclear factor kappa B (NF-kB) pathway [70].

MicroRNAs (miRNAs), a class of highly conserved, non-coding short (19–25 nucleotides in length) ribonucleic acids (RNAs), have been established as important regulators of gene expression. A number of miRNAs have been identified as regulators of fibrosis through their direct or indirect effects on the expression of genes encoding ECM components (reviewed in [71]). Studies have illustrated that EA preserved cardiac function and reduced fibrosis following myocardial infarction in a rat model [72]. Studies in isolated cells illustrated that these effects of EA were likely due to its up-regulation of miR-140-39, which in turn decreased the expression of mitogen-activated protein kinase kinase 6 (MKK6). Treatment of hepatoblastoma cell line (HepG2) with high glucose elicits oxidative stress and has been utilized as an in vitro model of diabetes. Treatment with EA mitigated high glucose-induced oxidative stress in these cells [73]. Mechanistically, this effect was dependent on the up-regulation of miR-133 by EA. miR-133 in turn down-regulated the expression of Kelch-like ECH-associated protein 1 (Keap 1), allowing nuclear factor erythroid 2-related factor (Nrf2) translocation to the nucleus and enhanced expression of antioxidant proteins.

### 2.4. Epigallocatechin-3-Gallate

Epigallocatechin-3-gallate (EGCG) is a natural phenolic compound, classified as a secondary metabolite, commonly found in various plant species (reviewed in [74]). Catechins, a subgroup of secondary metabolites, are widely present in tea plants. EGCG, a major catechin, is synthesized through specific branches of the flavonoid biosynthesis pathway. It possesses robust antioxidant and antibacterial capabilities. Nevertheless, these compounds suffer from drawbacks such as poor solubility, limited bioavailability under physiological conditions, and potential side effects in patients. Novel strategies are needed to address these challenges [74,75,76].

A number of studies have illustrated the anti-fibrotic effects of EGCG in the liver [77], kidney [78], lungs [79], heart [80] and other organs. Treatment with EGCG in isolated uterine fibroid or myometrial cells resulted in the reduction in mRNA or protein levels of critical fibrotic markers including fibronectin, collagen I, plasminogen activator inhibitor-1 (PAI-1), CTGF and α-SMA [81]. These studies also illustrated that EGCG altered the activation of JNK, Akt and β-catenin but interestingly not the Smad2/3 pathway. In contrast, the application of EGCG in a rat model of atrial fibrosis resulted in attenuated Smad2/3 activation in addition to diminished fibrosis and expression of collagens type I and III [82]. EGCG also attenuated lysyl oxidase expression, an enzyme involved in cross-linking of ECM proteins collagen and fibronectin. Studies with TGF-β-treated human umbilical vein endothelial cells (HUVEC), and miRNA microarray analysis indicated that EGCG enhanced the expression of miR-6757-3p in extracellular vesicles from these cells [83]. Treatment of human lung fibroblasts with extracellular vesicles derived from EGCG-treated HUVECs downregulated the expression of TGF-β receptor 1, as well as the levels of fibrosis-related genes, including fibronectin and α-SMA [83].

Exposure of endothelial cells to specific cytokines and/or growth factors including TGF-β and interleukin-1β results in the transition of these cells into a mesenchymal phenotype, a process termed endothelial-to-mesenchymal transition (EndMT). This provides one potential source of myofibroblasts in fibrotic conditions. Studies with HUVECs in vitro illustrated that EGCG is capable of attenuating cytokine-induced EndMT [84]. These studies also illustrated that EGCG diminished cytokine-induced expression of RhoA, production of ROS and activation of NF-kB and Smad signaling pathways. Studies by Zhang and colleagues [85] illustrated that ECGC directly binds to and activates PKCα. Computational docking analyses revealed that EGCG allosterically activated PKCα through interactions with the catalytic C3–C4 domain of this protein. In addition, an in vivo study involving type 2 diabetic rats showed that an eight-month treatment with EGCG reduced cardiac fibrosis [86]. This reduction occurred through the activation of autophagy by modulation of the AMP-activated protein kinase (AMPK)/mTOR pathway and subsequent suppression of the TGF-β pathway and MMPs.

### 2.5. Resveratrol

Resveratrol (3,4′,5-trihydroxystilbene) is a naturally occurring polyphenolic compound synthesized by plants in response to environmental stress. It is one of the most widely investigated bioactive compounds from plants and its notable attributes include antioxidant, anti-inflammatory, anti-cancer, cardiovascular-protective and anti-aging properties [87,88,89,90,91]. This compound is present in a number of plants including grapes, blueberries and peanuts (Table 2). It is efficiently absorbed and undergoes rapid and extensive metabolism within the body [87].

Numerous in vitro and in vivo studies have highlighted the ability of resveratrol to attenuate fibrosis via inhibition of the canonical (Smad 2/3) and non-canonical TGF-β signaling pathways [92,93,94,95,96]. An investigation conducted on rat lung tissue revealed that the introduction of resveratrol led to a significant downregulation in the protein and RNA expression of TGF-β1, α-SMA, Smad3/4, p-Smad3/4, CTGF, collagen and p-ERK1/2 [93]. Guo et al. described the cardiovascular protective effects of resveratrol, primarily attributed to its modulation of the non-canonical TGF-β/ERK1/2 signaling pathway [95]. Resveratrol treatment of isolated pterygium fibroblasts inhibited TGF-β1-induced proliferation, migration, contractility and expression of α-SMA, fibronectin and collagen type I [92]. Resveratrol treatment attenuated the activation of non-canonical p38 MAPK and PI3K/Akt pathways in these studies. Additionally, Wang and colleagues reported that, aside from downregulating TGF-β1/Smad3, resveratrol alleviated pulmonary fibrosis by downregulating lipopolysaccharide (LPS)/toll-Like Receptor 4 (TLR4)/Nuclear factor kappa B (NF-κB) signaling pathways in rats [94].

In murine models of cardiac hypertrophy, a study conducted by Ma and colleagues revealed that the administration of resveratrol significantly impeded Ang II-induced cardiac hypertrophy [97]. Moreover, resveratrol robustly alleviated Ang II-induced cardiac fibrosis and cardiac dysfunction. Furthermore, resveratrol demonstrated the ability to directly impede Ang II/AT1R signal transduction and prevent the Ang II-induced expression of pro-inflammatory cytokines, along with the activation of the NF-κB signaling pathway.

### 2.6. Genistein

Genistein is a polyphenolic isoflavone that belongs to the flavonoid group of plant compounds. As a group, the flavonoids play integral roles in signaling, auxin transport, pigmentation processes and conferring resistance to various stresses in plants. Given their disease-protective mechanisms, genistein and other flavonoids have the potential for extensive applications in disease prevention and treatment (reviewed in [98,99,100]). Genistein is naturally present in crucial food crops like soybeans and chickpeas. As a phytoestrogen, genistein structurally resembles estrogen, allowing it to mimic or counteract estrogenic effects [101]. Its potential to promote health spans various areas, showing promise in addressing complex conditions such as cancer, diabetes, cardiovascular diseases, Alzheimer’s and others [98,100]. In natural sources, isoflavones are found predominantly in glycosylated forms and become biologically active aglycones only after food processing. Pharmacokinetic studies have indicated the challenge of low oral bioavailability for genistein [102]. Studies on portal vein plasma levels have demonstrated that the bioavailability of genistein is higher in its aglycone form than in its glycoside form [103]. Another study highlighted that oral bioavailability is greater for genistein compared to genistin, the 7-*O*-beta-d-glucoside form of genistein and the predominant natural form of this isoflavone in plants [104].

Genistein has been shown to attenuate renal and cardiac fibrosis in diabetic rat models [105,106]. In both studies, treatment with genistein resulted in decreased expression of pro-inflammatory and pro-fibrotic cytokines including TGF-β, IL-1β and IL-6 as well as reduced Smad activation. Genistein increased the expression of Nrf2 and heme oxygenase-1 (HO-1) and conferred beneficial effects by protecting against oxidative injury, regulating apoptosis, modulating inflammation, and contributing to angiogenesis. The increased expression of Nrf2, HO-1 and NAD(P)H quinone oxidoreductase (NQO1) was speculated to provide the mechanism for reduced fibrotic and pro-inflammatory markers [106].

Another diabetes study revealed that genistein exhibits a clear anti-fibrotic impact on kidneys, both in an in vivo unilateral ureteral occlusion (UOO) model and an in vitro model involving renal epithelial cells [107]. Genistein elevated the expression of renal alkB homolog 5, RNA demethylase (ALKBH5), correlating to decreased expression of the myofibroblast phenotypic markers α-SMA and the zinc finger transcriptional repressor SNAI1 (Snail), thereby mitigating UUO-induced renal fibrosis, leading to an improvement in renal health. Additionally, oral administration of genistein notably improved liver injury and reduced collagen deposition in a dimethylnitrosamine (DMN)-induced liver fibrosis rat model [108]. Genistein inhibited the expression of hepatic stellate cell activation markers α-SMA and collagen I, both in vivo and in an HSC line (LX2 cells) in vitro. In rats, genistein also significantly reduced expression of MMPs 2 and 9, enzymes involved in ECM degradation. These studies clearly illustrated that genistein has direct effects on hepatic stellate cells, preventing their conversion to a myofibroblast phenotype. Genistein also reduced inflammatory cell infiltration and alleviated mRNA expression levels of inflammatory cytokines IL-1β, IL-6, TNF-α and monocyte chemoattractant protein-1 (MCP-1) in the liver of DMN-treated rats, suggesting modulation of important interactions between inflammation and fibrosis. Furthermore, genistein attenuated the expression of p-Janus kinase (JAK2)/JAK2, p-signal transducer and activators of transcription (STAT)3/STAT3, and suppressor of cytokine signaling 3 (SOCS3) proteins (associated with inflammatory processes), both in animals and in isolated cells [108].

Genistein has been indicated to have anti-fibrotic effects in in vivo and in vitro lung fibrosis models [109]. In studies with isolated cells, genistein attenuated TGF-β1-induced myofibroblast formation and expression of ECM components by inhibiting the activation of canonical and non-canonical TGF-β signaling pathways. As mentioned above, genistein has poor water solubility and low bioavailability in vivo. To address this, BIO 300 (Humaetics Corporation), a nanosuspension with genistein as the major active ingredient, has been developed and tested in several disease models [110]. In a mouse model of high-dose radiation-induced lung damage, treatment with BIO 300 improved mouse survival. Furthermore, treatment with BIO 300 preserved lung structure and reduced both inflammation and fibrosis.

### 2.7. Quercetin

Quercetin, a key bioflavonoid present in onions, apples, tea, brassica vegetables, nuts, seeds, bark, flowers and leaves, can also be extracted from various medicinal plants. It has been demonstrated to have therapeutic effects in wound healing, neuroprotection, vitiligo and other conditions [111,112,113,114]. A notable feature of quercetin is its structural abundance of OH groups (Table 2), which enables it to bind to ROS, preserving cell viability and exhibiting a higher antioxidant capacity than many other flavonoids [115]. However, its limited water solubility poses a hurdle for effective treatment. Additionally, quercetin faces low bioavailability in the body due to challenges in absorption, rapid metabolism and swift elimination. To address these issues and enhance the stability and bioavailability of quercetin, researchers have endeavored to chemically modify it and related flavonoids, aiming to improve aqueous solubility while retaining bioactivity (reviewed in [116]).

Numerous studies with isolated cells and animal models have illustrated the anti-fibrotic effects of quercetin [117,118,119]. Quercetin modulates fibrosis through multiple cellular and molecular mechanisms and pathways. A study conducted by Xiao and colleagues [120] demonstrated that one of the mechanisms by which quercetin exerts its anti-fibrotic effects is by attenuating the TGF-β/AKT/mTOR signaling pathway. They also found that a human embryonic lung fibroblast cell line pretreated with quercetin exhibited decreased expression of proinflammatory and profibrotic factors such as IL-6, IL-8, collagen type I and collagen type III. In the same study, it was shown that in vivo nasogastric administration of quercetin reversed the increase in profibrotic factors, including epidermal growth factor, IL-6, TGF-β, collagen I and collagen III, as well as fibrotic signaling mediators mTOR and AKT, in a rabbit tracheal stenosis model. Another study in a mouse model of silicosis and isolated murine alveolar macrophages showed that quercetin treatment alleviated pulmonary fibrosis by inhibiting macrophage-to-myofibroblast transition through interference in the TGF-β-Smad2/3 signaling pathway [121]. A study conducted on human gingival fibroblasts demonstrated that pre-treatment with quercetin activated PPAR-γ, and subsequently suppressed NF-κB and attenuated the production of pro-fibrotic cytokines IL-1β, IL-6, IL-8 and TNF-α [122].

Cyclophosphamide treatment results in substantial structural damage in the liver and fibrosis in rats. Treatment of rats with quercetin for five days prior to treatment with cyclophosphamide protected the liver from structural damage and significantly suppressed fibrosis [123]. Interestingly, treatment of animals with quercetin after the onset of fibrosis reversed structural damage and fibrosis similar to the levels seen in animals treated prior to cyclophosphamide exposure solidifying its therapeutic efficacy. Studies such as these are increasingly being incorporated into experimental design to distinguish between prevention and reversal effects on fibrosis [123].

Using a mouse unilateral ureteral obstruction model, Lu and colleagues [124] illustrated that quercetin treatment preserved kidney function, reduced the expression of inflammatory markers and prevented interstitial damage and fibrosis. Furthermore, quercetin treatment inhibited macrophage infiltration and the polarization of macrophages to both M1 and M2 phenotypes likely through antagonism of NF-kB signaling and inactivation of the canonical TGF-β/Smad2/3 pathway. These studies support important interactions between inflammatory/immune cells and modulation of fibrosis that can potentially be targeted by plant-derived compounds.

Fibrosis is associated with cellular senescence or arrested cell growth in some tissues [125,126]. The biological function of cellular senescence is complex, being reported as beneficial and deleterious, probably depending on the pathophysiological context. It is presently recognized as a pivotal pathogenic element in pulmonary fibrosis [127]. In aged mice, apoptosis-resistant senescent fibroblasts promote ongoing lung fibrosis in a bleomycin model [128]. Treatment of fibroblasts from the lungs of idiopathic pulmonary fibrosis patients with quercetin mitigated apoptosis resistance of these cells in part via inactivation of Akt and upregulation of the FasL receptor [127]. Recent studies have taken advantage of an engineered heart-on-a-chip model to evaluate the association between cellular senescence and cardiac fibrosis [129]. For these studies, the heart-on-a-chip model was created by the inclusion of human cardiac fibroblasts and cardiomyocytes derived from human induced pluripotent stem cells. Treatment with TGF-β induced a profibrotic phenotype as well as expression of senescence markers including cyclin-dependent kinase inhibitor (CDKN)1A, CDKN1B, Serpin Family E Member 1 (SERPINE1), among others. Treatment of the heart-on-a-chip with the senolytic drug combination, dasatinib and quercetin (DQ), reduced the expression of senescence markers and improved contractile function. While the treatment reduced the relative proportion of fibroblasts, it did not alter collagen deposition. Studies have also been carried out with this senolytic combination in patients with idiopathic pulmonary fibrosis [130]. In a small cohort of patients, treatment with DQ significantly improved physical function including gait speed and walking distance. Further, DQ treatment showed a trend to reduce several senescence-associated secretory products including matrix-modifying enzymes, MMP-7 in particular. However, the patient population was too small for meaningful conclusions. Understanding the role of cellular senescence in organ fibrosis warrants further investigation as this may provide novel therapeutic opportunities.

### 2.8. Naringin/Naringenin

Naringin is a flavonoid primarily found in citrus fruits like grapefruit where it is responsible for the fruit’s bitter taste. Naringin is metabolized in the gut by naringinase into a flavorless and water-insoluble active compound, naringenin (reviewed in [131]). With a diverse pharmacological profile encompassing anticancer, anti-inflammatory and antioxidant properties, naringenin has demonstrated considerable therapeutic potential (reviewed in [132,133,134]). Due to its low solubility, effective techniques have been employed to enhance oral bioavailability, notably through the utilization of nanotechnologies [135,136]. Despite this challenge, data obtained from in vitro and in vivo studies consistently reveal favorable effects of pure naringenin, naringenin-loaded nanoparticles and combinations of naringenin with other bioactive compounds.

Studies in isolated cells and animal models indicate that naringin can modulate the fibrotic response in multiple organs including the liver [137], kidney [138], lung [139] and heart [140]. In many cases, this appears to be due to the antioxidant and anti-inflammatory capabilities of naringin. Meng and colleagues reported that naringin treatment interferes with the TGF-β/Smad2/3 signaling pathway [138], a finding supported by other studies [141,142]. It was also demonstrated that naringenin reversed liver injury by normalizing elevated markers of biochemical and oxidative stress, as well as fibrosis. Furthermore, it restored the normal activity of MMP-9 and MMP-2, and also prevented an increase in protein levels of NF-kB, IL-1β, IL-10, TGF-β, CTGF, Collagen I, MMP-13 and Smad7 [142].

Qin and colleagues recently developed a zebrafish model in which treatment with thioacetamide induces substantial liver fibrosis and steatosis [143]. Treatment with naringin attenuated both liver fibrosis and steatosis in this model. Similar to previous studies with naringin [144], treatment of zebrafish with this compound increased Nrf2 and antioxidant enzyme levels and reduced the production of inflammatory cytokines. Morpholino experiments suggested that indoleamine 2,3-dioxygenase 1 (IDO1) may, at least in part, mediate the effects of naringin in the zebrafish model. This heme-containing enzyme is involved in tryptophan metabolism and has been shown to have immunosuppressive and anti-inflammatory effects [145]. Other studies have demonstrated that treatment with naringenin can activate the PPAR-γ pathway, thereby suppressing NF-kB and the inflammatory response including the release of cytokines that may induce fibrosis [146,147,148].

Studies are beginning to take advantage of various modeling approaches to predict molecular mechanisms underlying the effects of plant-derived compounds in diverse diseases. These approaches can rapidly identify potential signaling pathways impacted by these compounds and predict their therapeutic efficacy and potential side effects. Recent studies have utilized network pharmacology and molecular docking to help reveal mechanisms by which naringin prevents renal fibrosis [149]. After database screening, over two hundred common targets were identified for naringin and renal fibrosis. Molecular docking analysis indicated Akt effectively interacts with naringin. The PI3K/Akt pathway was experimentally determined to play an important role in the effects of naringin on the inhibition of renal fibrosis using isolated cells and an animal model. These studies provide a proof-of-concept that should form a foundation for other studies evaluating mechanisms of the anti-fibrotic effects of plant-derived compounds.

### 2.9. Sulforaphane

Sulforaphane (SFN) is an isothiocyanate present in cruciferous vegetables like broccoli and brussel sprouts (Table 2). It is derived from glucoraphanin and is produced through the activity of myrosinase, a β-thioglucosidase found in either plant tissue or the mammalian microbiome. Studies indicate that the administration of SFN leads to significant excretion in urine, and this remains stable over time, especially in aqueous solutions [150,151]. Conversely, the use of dried extracts from broccoli sprouts or seeds that are rich in glucoraphanin (GR), the primary glucosinolate in broccoli and broccoli sprouts, results in a highly variable conversion of GR to SFN metabolites. Administering GR in this form, especially in combination with other molecules like myrosinase, significantly enhances its bioavailability [150].

SFN has been indicated to have chemoprotective effects in part through induction of cell cycle arrest and apoptosis [152,153]. It has also been reported to inhibit epithelial-to-mesenchymal transition (EMT), an important process in cancer metastasis and fibrosis [154,155,156]. SFN has been shown to have anti-fibrotic effects in isolated cells and animal models [156,157] and to promote the dedifferentiation of myofibroblasts [158]. SFN exerts an anti-fibrotic effect through intricate networks and multiple signaling pathways. For example, one of the mechanisms described is through downregulation of the canonical and non-canonical TGF-β signaling pathways, with subsequent diminution of fibrogenic molecules [159,160,161]. In a radiation-induced skeletal muscle fibrosis model, SFN treatment reserved muscle fiber organization, reduced collagen fiber content, and mitigated the infiltration of inflammatory cells [160]. Additionally, SFN triggered the upregulation and activation of Nrf2, activation of Akt and the suppression of glycogen synthase kinase-3 beta (GSK-3β) and Fyn accumulation. GSK-3β is a serine/threonine protein kinase that regulates glycogen metabolism. Activation of the GSK-3β/Fyn pathway participates in the export and degradation of nuclear Nrf2 resulting in enhanced oxidative stress. Activation of GSK-3β has been demonstrated to enhance fibrosis via multiple pathways leading to its implication as a potential therapeutic target for fibrosis [161,162].

A study conducted by Ishida and colleagues [163] using a mouse model of liver fibrosis (alcohol plus carbon tetrachloride) revealed that SFN treatment inhibited Kupffer cell infiltration, fibrosis, fat accumulation and lipid peroxidation. SFN also exerted an anti-inflammatory effect that was likely due to attenuated toll-like receptor 4 (TLR4) signaling. Studies in liver cell lines suggested the upregulation of Nrf2-regulated antioxidant genes, such as heme oxygenase 1 (HMOX1), NQO1 and glutathione S-transferase mu 3 (GSTM3) as the underlying mechanisms of the anti-fibrotic effects of SFN. Another study showed that an SFN-enriched diet inhibited the diabetes-induced rise in histone deacetylase 2 (HDAC2) activity which correlated with histone acetylation and the transcriptional activation of the bone morphogenetic protein (BMP)-7 promoter (related to TGF-β/Smad signaling) [164]. Additional studies indicate that one of the anti-fibrotic mechanisms of SFN involves the inhibition of Ang II signaling, leading to the upregulation of Nrf2 and its downstream antioxidant genes [165]. In addition, SFN averted Ang II-induced aortic damage by activating Nrf2 through the ERK/GSK-3β/FYN Proto-Oncogene, Src Family Tyrosine Kinase (Fyn) pathway [166].

### 2.10. α-Lipoic Acid

Alpha-lipoic acid (ALA), also known as thioctic acid, is a naturally occurring essential organosulfur dithiol compound derived from plants, animals, and humans [167,168]. Its role as a nutritional cofactor for mitochondrial enzymes is pivotal in glucose and energy metabolism [167]. Despite its significant antioxidant potential, the oral bioavailability of ALA is limited by pharmacokinetic factors such as reduced solubility, lack of gastric stability and hepatic degradation [167,169]. These limitations are further compounded by formulation challenges. ALA, being a potent direct free-radical scavenger, has the capability to regenerate other endogenous antioxidants, contributing to its overall antioxidant efficacy [167].

Similar to a number of other plant-derived compounds, studies have demonstrated that one of the mechanisms through which ALA exerts its anti-fibrotic activity is by inhibiting the TGF-β/Smad2/3 signaling pathway [170,171]. In a study conducted by Ibrahim and colleagues, male Wistar rats were subjected to amiodarone, an anti-arrhythmic drug that induces pulmonary fibrosis. Treatment with ALA significantly prevented amiodarone-induced lung damage, fibrosis, oxidative stress and inflammation [170]. Furthermore, ALA treatment also resulted in decreased serum levels of TGF-β and IFN-γ, a pro-fibrotic factor that exhibits stage-specific participation in pulmonary inflammation. A reduction in α-SMA, collagen I and fibronectin by ALA was also described in a mouse model of bleomycin-induced scleroderma [171]. These studies illustrated that ALA treatment ameliorated bleomycin-induced Nox4 and p22phox expression and total oxidant status.

Studies also demonstrated that in fibroblasts treated with ALA and H_2_O_2_, oxidative stress and inflammation biomarkers such as TNF-α, IL-1β and IL-6, along with NF-kB, were significantly lower compared to cells exposed to H_2_O_2_ only [172]. These findings are supported by another study conducted by Yoon and colleagues, where pretreatment of fibroblasts with ALA inhibited IL-1β-induced activation of NF-kB, reduced IL-8 expression, and mitigated mitochondrial dysfunction by lowering levels of ROS [173].

### 2.11. Emodin

Emodin (1,3,8-trihydroxy-6-methylanthraquinone) is an anthraquinone derivative that is found in several plants and is thought to be the primary pharmacodynamic component of rhubarb. Emodin possesses an array of biological activities including modulation of immune and inflammatory processes, suppression of viral infection, protection against neurological damage, and attenuation of tumor growth and metastasis [174,175]. These properties provide the basis for the therapeutic utilization of emodin in a variety of pathological conditions including glaucoma, ulcerative colitis, cancer, Alzheimer’s disease, sepsis, malaria, inflammatory bowel disease and others. Emodin’s anti-cancer effects are at least in part mediated by its inhibition of the TGF-β1-mediated crosstalk between tumor-associated macrophages and cancer cells [176,177]. Concern has been expressed that prolonged use of emodin at high doses may initiate adverse effects including hepatotoxicity although the mechanisms of these effects are not well-defined [178] and most studies show no or neglectable toxicity of emodin [179].

Emodin has also been shown to suppress fibrosis in several in vitro and in vivo models (reviewed in [180]). Treatment of isolated cells and animal models with emodin attenuated the activation of canonical Smad and non-canonical (p38 MAPK) TGF-β signaling pathways [181,182,183]. Similarly, in a rat model in which liver fibrosis was induced by carbon tetrachloride, treatment with emodin repressed expression of TGF-β1, Slug and Snail, all involved in the formation of myofibroblasts and fibrosis [184]. In an in vivo silica inhalation-induced model of pulmonary fibrosis, Sirt1 expression was increased following treatment with emodin, which led to Smad3 deacetylation and inhibition of TGF-B/Smad2/3 signaling [185].

As mentioned above, emodin can modulate inflammatory and immune cell function and may thus impact fibrosis indirectly. Several studies have illustrated that emodin can suppress NF-kB pathway activation and inflammatory cytokine production [186,187]. In lipopolysaccharide (LPS)-treated human keratinocytes (HaCaT cells), emodin prevented the LPS-induced suppression of mir-21 [188]. This resulted, directly or indirectly, in decreased activation of NF-kB signaling, enhanced activation of PI3K/Akt and decreased production of inflammatory mediators. Inhibition of NF-kB signaling by emodin attenuated M1 polarization of macrophages while M2 polarization was activated by suppression of interferon regulatory factor (IRF)4/STAT6 signaling [177]. Formation of M2 macrophages has been thought to be beneficial in the resolution of inflammatory responses; however, this process is likely more complex and the role of emodin-induced macrophage polarization in fibrosis requires further exploration.

## 3. Conclusions

The impact of fibrosis on tissue and organ function is increasingly being appreciated. Despite this, few treatments have been approved specifically for fibrosis. The complex pathways underlying fibrosis and the utilization of these pathways for other physiological processes result in many compounds having “off-target” effects and deleteriously impacting important cellular functions. Plant-derived compounds are aggressively being pursued as alternative treatments for fibrosis given their broad pleiotropic effects and subsequent ability to target the complex cellular and molecular mechanisms driving fibrosis. Indeed, studies in isolated cells and animal models have demonstrated the potential of select plant-derived compounds to protect against and in some cases reverse organ fibrosis with no or only minor side effects.

Despite the successes of some plant-derived natural compounds in attenuating fibrosis in preclinical models, a number of hurdles remain prior to their widespread clinical utilization. Identification of underlying cellular and molecular mechanisms of these compounds will assist in determining the specificity of compounds in targeting fibrosis and elucidation of potential “off-target” effects. Exciting research utilizing molecular docking approaches is helping identify specific molecular mechanisms of these compounds. Studies utilizing computational modeling approaches to interrogate signaling pathways involved in fibrosis [189] are accelerating the identification of underlying mechanisms of plant-derived compounds. While relatively safe, some plant-derived compounds have been illustrated to exhibit cytotoxicity, particularly at high doses. Most studies to date have been carried out in rodent models of fibrosis and research in larger animal models with physiological parameters more similar to humans will be important to evaluate pharmacokinetics and side effects of plant-derived compounds. A number of these compounds have poor bioavailability and novel approaches are being developed to modify and/or deliver these compounds to address this.

**Table 1 cells-13-00421-t001:** Anti-fibrotic mechanisms of major plant-derived compounds.

Compound	Mechanisms of Anti-Fibrotic Effects	References
Curcumin	- Suppress canonical TGF-β1/SMAD2/3 signaling - Inhibit the TAK1 pathway - Modulate macrophage-fibroblast crosstalk - Inhibit IL18 expression - Modulate Ang II receptor expression - Reduce blood pressure - Increase PPAR-γ activity - Reduce PKC activity	[29,30,31,32,33,34,35,36,37,38,39,40,45,46,47,48,49,50]
Capsaicin	- Inhibit canonical TGF-β1/Smad2/3 signaling - Prevent the activation of macrophages - Attenuate inflammatory cytokine production (TNF-α) - Upregulate PPARγ signaling - Inhibit ERK1/2/eIF3a signaling pathway	[51,52,53,54,55,56,57,58]
Ellagic acid	- Decrease the canonical TGF-β/Smad2/3 signaling - Increase antioxidant protein production - Decrease expression of inflammatory cytokines (IL-1β, IL-6, and TNF-α) - Attenuate the activation of JAK2/STAT3 pathway - Increase PPAR-γ signaling - Decrease ACE 1 expression and Ang II formation - Increase miR-133 expression (increasing Nrf2 translocation to the nucleus)	[59,60,61,62,63,64,66,67,68,69,70,71,72,73]
Epigallocatechin-3-gallate (egcg)	- Diminish collagen synthesis and LOX expression - Downregulate expression of TGFBR1, fibronectin and α-SMA - Increase miR-6757-3p - Decrease endothelial-to-mesenchymal transition (EndMT) process and myofibroblast formation - Activate PKCα - Modulate the AMPK/mTOR pathway.	[74,75,76,77,78,79,80,81,82,83,84,85,86]
Resveratrol	- Inhibit canonical TGF-β1/Smad2/3 signaling - Downregulate non-canonical TGF-β/ERK1/2 and p38 MAPK pathways - Downregulate TLR4/NF-κB signaling pathways - Attenuate PI3K/AKT pathways. - Inhibit Ang II/AT1R signal transduction and prevents the Ang II-induced expressions of pro-inflammatory cytokines.	[87,88,89,90,91,92,93,94,95,96,97]
Genistein	- Decrease activation of canonical TGF-β/Smad2/3 pathway - Increase antioxidant protein production - Elevate the expression of ALKBH5. - Inhibit formation of myofibroblasts - Decrease expression of MMP2/9 and TIMP1 - Attenuate expression proinflammatory cytokines (IL-1β, IL-6, TNF-α, and MCP-1). - Attenuate activation of JAK2/STAT3 pathway - Increase the expression of ER-β and PPAR-γ	[98,99,100,101,102,103,104,105,106,107,108,109,110]
Quercetin	- Attenuate the TGF-β/AKT/mTOR signaling - Decrease proinflammatory and profibrotic protein expression - Inhibit macrophage-to-myofibroblast transition. - Activate PPAR-γ, suppressing NF-κB and attenuating the production of IL-1β, IL-6, IL-8, and TNF-α.	[111,112,113,114,115,116,117,118,119,120,121,122,123,124,127]
Naringenin	- Attenuate canonical TGF-β/Smad2/3 signaling - Inhibit AngII-induced proliferation of fibroblasts - Decrease expression of α-SMA, collagen 1, collagen 3, IL-1β, IL-6, and TNF-α. - Activate the PPAR-γ pathway	[131,132,133,134,135,136,137,138,139,140,141,142,143,144,146,147,148,149]
Sulforaphane	- Attenuate canonical TGF-β1/Smad2/3 signaling - Reduce collagen fiber content - Upregulate Nrf2 antioxidant proteins - Activate AKT and suppress GSK-3β and Fyn accumulation - Reduce reactive oxygen via downregulation of NOX1 and NOX4 - Attenuate LPS/toll-like receptor 4-mediated sensitization to TGF-β - Inhibit AngII expression	[150,151,152,153,154,155,156,157,158,159,160,163,164,165,166]
α-Lipoic acid	- Inhibit canonical TGF-β/Smad2/3 signaling pathway - Decrease ECM expression and prevent collagen deposition - Decrease mRNA expression levels of IFN-γ - Reduce oxidative stress - Decrease NF-κB activation and inflammatory cytokine expression	[167,168,169,170,171,172,173]
Emodin	- Attenuate canonical and non-canonical TGF-β (p38 MAPK) signaling pathways - Decrease NF-kB activation - Increase PI3K/Akt signaling	[174,175,176,177,178,179,180,181,182,183,184,185,186,187,188]

**Table 2 cells-13-00421-t002:** Chemical structures and primary countries of origin of plant-derived compounds discussed in this review. Chemical structures were obtained from the National Library of Medicine, PubChem database.

Plant-Derived Compound	Structure	Origin	Reference
Curcumin	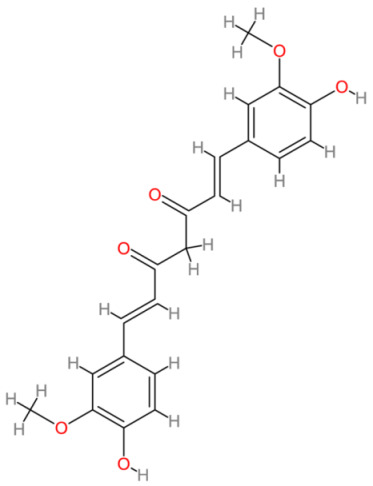	Asia	[190,191]
Capsaicin	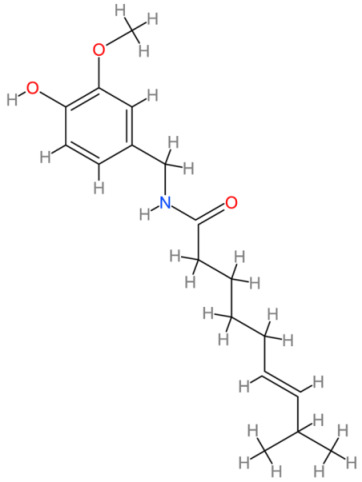	Americas	[192,193]
Ellagic acid	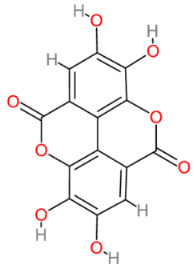	Multinational	[194,195]
Epigallocatechin gallate	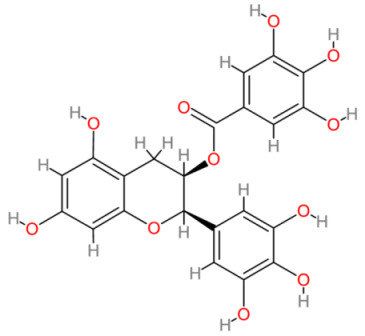	Multinational	[196,197]
Resveratrol	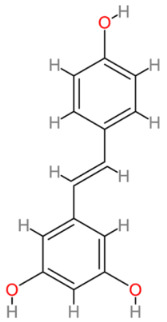	China and Japan	[198,199]
Genistein	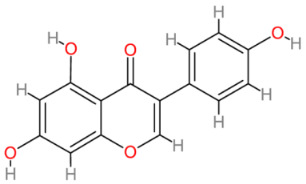	Multinational	[200,201]
Quercetin	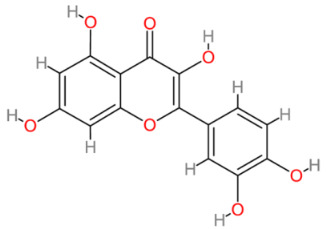	Multinational	[202,203]
Naringenin	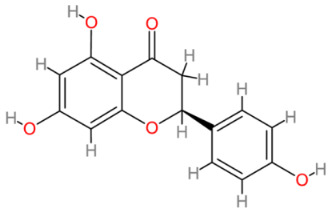	Multinational	[204,205]
Sulforaphane	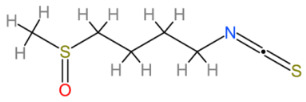	Multinational	[206,207]
α-Lipoic acid	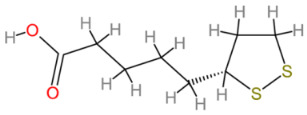	Multinational	[208,209]
Emodin	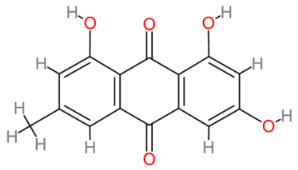	Multinational	[175,210]

## Figures and Tables

**Figure 1 cells-13-00421-f001:**
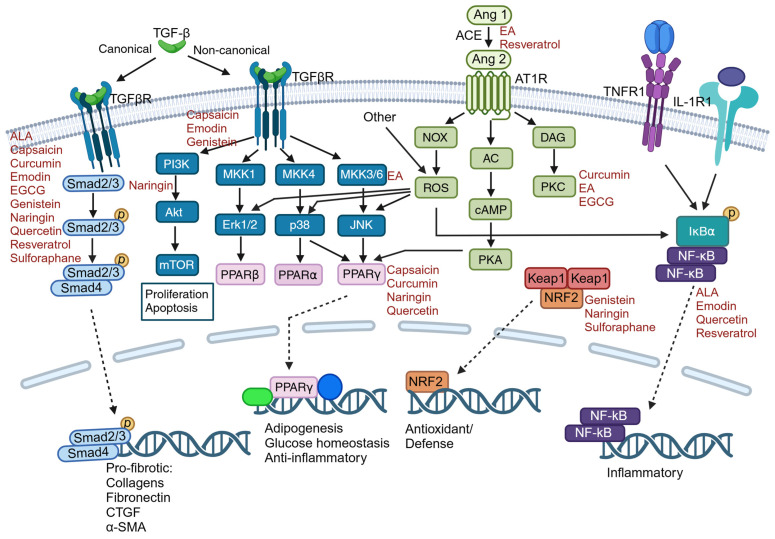
Schematic of major fibrosis pathways impacted by common plant-derived compounds (red font) studied to date. Pathways and compounds that modulate them are discussed in detail in the following sections and summarized in Table 1. (ALA, α-lipoic acid; EGCG, Epigallocatechin-3-gallate; TGF-β, transforming growth factor-β; Ang, angiotensin; PI3K, phosphoinositide-3 kinase; and EA, epigallic acid). Created with BioRender.com, accessed on 14 January 2024.

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
