# Peer review of "Potential of Plant-Derived Compounds in Preventing and Reversing Organ Fibrosis and the Underlying Mechanisms"

_cells, 2024, doi:10.3390/cells13050421_

Round 1

Reviewer 1 Report

Comments and Suggestions for Authors

Azeredo et al have looked at the Potential of plant-derived compounds in preventing and reversing organ fibrosis and the underlying mechanisms and it is a welcome idea. However, some points below should be clarified /noted

There are some statements made that needs appropriate references. Such as 

Line 154/155; 177/178

Some acronyms were used and was not provided in full at first mention e.g line 328 ...PKCα and few others. Authors should do the needful

In describing the sources of the active polyphenols, authors should endeavour to state the major country source of the ingredients/spices from where this active polyphenols are gotten. Also, I encourage putting structures of these compounds in the manuscript. All can be in a figure or placed accordingly following the discussion pattern.

Moderate English editing of the manuscript is recommended.

Comments on the Quality of English Language

Moderate English editing of the manuscript is recommended

Author Response

The authors would like to thank the reviewer for careful consideration of this manuscript and for their insightful suggestions.  Below we summarize the modifications made to the manuscript.  The modifications are highlighted in the revised submission.

  • There are some statements made that needs appropriate references.

Thank you for pointing this out.  In our attempt to keep the number of references at a minimum, we failed to cite several key statements.  We have now added references to these statements.  Specifically what was lines 154-163 are now appropriately cited as are lines 177-180.

  • Some acronyms were used and were not provided in full at first mention..

Our apologies for not spelling out some of the acronyms at first mention in the text.  We have corrected this.

  • …authors should endeavor to state the major country source…

Thank you for this suggestion.  We have added a second table to the manuscript that includes the chemical structure of the compounds discussed as well as the major country(ies) of origin of the compounds.

  • We have also carefully edited the manuscript for grammatical errors.

Reviewer 2 Report

Comments and Suggestions for Authors

There are several issues that need to be fixed with the references. These problems require work on their part and repeat throughout the manuscript.

All of the similar problems throughout the manuscript (not just the ones I am identifying) need to be fixed.

Ref 1 is a review – I think it you should refer to a review as “reviewed in 1” (this applies to every relevant ref in the manuscript), otherwise the reader does not know if you are taking us to the source of the research or not – and that is a really important part of a review.

Ref 2 is a study in mice. It is important that you state the species in the text, otherwise the reader will be misled into thinking that a body of work exists in humans that does not: e.g. “for instance, in a murine model, developmental deficits in….”

Ref 5. As far as I can see, this paper does not contain data showing that “fibrosis… contributes to up to 45% of deaths in the United States annually”. Indeed, ref 5 is a review anyway, so it is not satisfactory to make this claim, even if they did state it – because this sort of claim should go back to the original source research data/publication – otherwise we end up with reviews of reviews, quoting papers and data that may not exist.

The paragraph starting at line 60 has many claims that are not backed up with references

Ref 23: this is an important claim, but only one paper is referenced, and this study has an n=2. Are there other studies to support this claim?

Lines 149-155 are missing references

Lines 164-170 are missing references

Lines 172-178 are missing references

Ref 33: this interesting study shows that curcumin could modulate PKC – both activating and inhibiting it depending on Ca+, but that’s not how you describe the study.

Author Response

The authors would like to thank the reviewer for careful consideration of this manuscript and for their insightful suggestions.  Below we summarize the modifications made to the manuscript.  The modifications are highlighted in the revised submission.

  • Ref 1 is a review – I think you should refer to a review as “reviewed in 1”…

This is an excellent suggestion.  We have eliminated some of the citations to review articles and rely more heavily on primary research articles.  For citations to reviews, we now indicate that the citation is referring to a review article.

  • Ref 2 is a study in mice….

This is an excellent suggestion and helps put the literature in context.  The suggested wording has been added to the discussion of this study and several others involving animal models or isolated cells.

  • Ref 5. As far as I can see, this paper does not contain data showing that “fibrosis…contributes up to 45% of deaths in the United States annually”.

We have clarified this statement and provide a citation that analyzed causes of death from  the 2019 Global Burden of Disease study and found that fibrosis contributed to between 17.8 and 35.4% of deaths worldwide in 2019. 

  • Thank you for pointing out statements that require citations. In our efforts to minimize the number of references, we inadvertently omitted several references and failed to cite several key statements.  We have added references to statements indicated by the reviewers.

  • Ref 33: this interesting study shows that curcumin…

We did a poor job summarizing this study and have reworked the relevant portions of this paragraph (lines 193-198).

Round 2

Reviewer 2 Report

Comments and Suggestions for Authors

The authors have made the necessary improvements to the manuscript